# Adenosine Kinase Isoforms in the Developing Rat Hippocampus after LiCl/Pilocarpine Status Epilepticus

**DOI:** 10.3390/ijms23052510

**Published:** 2022-02-24

**Authors:** Petr Fábera, Libor Uttl, Hana Kubová, Grygoriy Tsenov, Pavel Mareš

**Affiliations:** 1Department of Developmental Epileptology, Institute of Physiology, Czech Academy of Sciences, 14200 Prague, Czech Republic; libor.kvary@seznam.cz (L.U.); hana.kubova@fgu.cas.cz (H.K.); grygoriy.tsenov@gmail.com (G.T.); pavel.mares@fgu.cas.cz (P.M.); 2Department of Neurology, Second Faculty of Medicine, Motol University Hospital, Charles University, 15006 Prague, Czech Republic; 3National Institute of Mental Health, 25067 Klecany, Czech Republic

**Keywords:** adenosine kinase, isoforms, inhibitor, development, hippocampus, epileptic afterdischarges, rat, LiCl/pilocarpine, status epilepticus

## Abstract

LiCl/pilocarpine status epilepticus (SE) induced in immature rats leads, after a latent period, to hippocampal hyperexcitability. The excitability may be influenced by adenosine, which exhibits anticonvulsant activity. The concentration of adenosine is regulated by adenosine kinase (ADK) present in two isoforms—ADK-L and ADK-S. The main goal of the study is to elucidate the changes in ADK isoform expression after LiCl/pilocarpine SE and whether potential changes, as well as inhibition of ADK by 5-iodotubercidin (5-ITU), may contribute to changes in hippocampal excitability during brain development. LiCl/pilocarpine SE was elicited in 12-day-old rats. Hippocampal excitability in immature rats was studied by the model of hippocampal afterdischarges (ADs), in which we demonstrated the potential inhibitory effect of 5-ITU. ADs demonstrated significantly decreased hippocampal excitability 3 days after SE induction, whereas significant hyperexcitability after 20 days compared to controls was shown. 5-ITU administration showed its inhibitory effect on the ADs in 32-day-old SE rats compared to SE rats without 5-ITU. Moreover, both ADK isoforms were examined in the immature rat hippocampus. The ADK-L isoform demonstrated significantly decreased expression in 12-day-old SE rats compared to the appropriate naïve rats, whereas increased ADK-S isoform expression was revealed. A decreasing ADK-L/-S ratio showed the declining dominance of ADK-L isoform during early brain development. LiCl/pilocarpine SE increased the excitability of the hippocampus 20 days after SE induction. The ADK inhibitor 5-ITU exhibited anticonvulsant activity at the same age. Age-related differences in hippocampal excitability after SE might correspond to the development of ADK isoform levels in the hippocampus.

## 1. Introduction

Adenosine is considered an endogenous homeostatic and metabolic neuromodulator that fulfills its functions through the activation of G-protein-coupled receptors, including A1, A2A, A2B, and A3 [1,2,3]. In the brain, adenosine acts as a seizure suppressor and seizure terminator, with the inhibitory effects mediated by both A1 and A2A receptors [4,5,6]. Adenosine is a crucial component of purinergic signaling and is tightly regulated by multiple enzymes and transporters [7]. Brain levels of adenosine are regulated primarily by the activity of adenosine kinase (ADK) [8], a key regulatory enzyme that catalyzes the phosphorylation of adenosine into 5′-adenosine-monophosphate (AMP) [9]. Slight changes in ADK expression may result in major changes in adenosine concentrations. The overexpression of ADK leads to a rapid decrease in extracellular adenosine [10,11,12]. In contrast, the downregulation or pharmacological inhibition of ADK leads to increased concentrations of adenosine, resulting in diminished excitability in the brain [13]. The manifestation of epileptic seizures may be associated with the dysfunction of adenosine-mediated inhibitory tone [14,15]. Therefore, changes in ADK expression become crucial for regulating adenosine tone and may contribute to the process of epileptogenesis [16,17].

ADK exists in two isoforms, i.e., long (ADK-L 40.5 kDa) and short (ADK-S 38.7 kDa), which are expressed in mammalian cells [18]. The long nuclear isoform ADK-L plays a role as an epigenetic regulator [19]. Epigenetic mechanisms, including DNA methylation, lead to altered gene expression and may induce neuronal excitability [20]. Increased DNA methylation contributes to seizure susceptibility and epilepsy development [21]. The ADK-L isoform is considered to be more effective in DNA methylation compared to the ADK-S isoform [19]. The short cytoplasmic isoform ADK-S regulates the extracellular tissue tone of adenosine and, thus, the degree of adenosine receptor activation [8]. The developmental representation of ADK isoforms is based on studies performed in the neocortex [22,23] and cerebellum. The specific role of these isoforms in the developing hippocampus must be elucidated.

In the adult brain, ADK is expressed mainly in astrocytes. The predominant astrocytic expression replaces a predominantly neuronal ADK expression present in early brain development [22,23]. In the hippocampus, the described changes occur during the first three postnatal weeks, with a major shift in ADK protein expression occurring toward the end of the first postnatal week [23,24]. Moreover, the switch from ADK-L toward ADK-S during early postnatal brain development has recently been described [18,23,25]. Temporal and spatial ADK isoform expression is highly regulated.

The function of ADK isoforms in the developing hippocampus is still unclear. In this study, we investigated the changes in ADK isoform expression during hippocampal development and whether potential changes may contribute to changes in hippocampal excitability after an insult during early brain development. Rats in developmental stages corresponding to the human perinatal period, preschool (infantile), school-age children (juvenile), and young adults (adolescent) were used [26] (Figure 1).

We studied the hippocampal afterdischarges (ADs) after LiCl/pilocarpine status epilepticus (SE). Electrical stimulation has been widely used in order to induce habitual seizures in epilepsy patients and study characteristics for ADs [27,28]. The epileptic AD model resembles focal temporal seizures if applied to limbic structures [29,30,31]. The most common target for electrical stimulation of AD in limbic structures is the hippocampus [32,33]. The hippocampus is highly susceptible to AD release and can appear in epileptogenic and nonepileptogenic brains [34].

LiCl/pilocarpine status epilepticus (SE) is the most common model of temporal lobe epilepsy [35]. We decided to study the LiCl/pilocarpine SE induced in 12-day-old rats where the development of epilepsy in adulthood has already been demonstrated in our laboratory [36]. In correlation to humans, a consistent finding across multiple studies is that the highest incidence of SE is among children under 1 year of age [37,38]. One human year almost is equal to approximately two rat weeks (13.8 days) when correlating their entire life span [39].

SE results in astrogliosis and an associated increase in ADK expression [22,40], but the role of the ADK isoforms in hippocampal excitability after LiCl/pilocarpine SE in the immature brain has not been elucidated.

We also studied the activity of a nonselective inhibitor of ADK, 5-iodotubercidin (5-ITU), on hippocampal ADs and its possible changes with age. 5-ITU has been demonstrated to increase extracellular adenosine levels in the rat brain [41] and may attenuate epilepsy development after SE in immature rodent brains [42].

The developmental profile of ADK isoforms, their function, and changes after brain insult might be crucial to identifying new potential therapeutic strategies in pediatric neurology.

## 2. Results

### 2.1. Threshold Intensity

The changes in the threshold intensity of hippocampal ADs of SE rats differed from those observed in control animals. Elicitation of hippocampal ADs in 15-day-old SE rats required significantly higher threshold intensities compared to age-matched controls (*p* = 0.0017; F = 17.50). An opposite effect was observed in P32 rats (*p* = 0.0417; F = 15.42). Data from 12, 18, 25, and 45-day-old rats did not demonstrate any significant differences in the AD threshold intensity between control and SE animals (Figure 2).

The inhibitor of ADK, 5-ITU, did not cause any changes in hippocampal AD threshold in 12-day-old SE rats compared to age-matched SE rats. In 5-ITU-pretreated 15-day- old SE rats, the elicitation of hippocampal ADs required significantly lower threshold intensities than age-matched SE rats (*p* = 0.0081; F = 19.74). In contrast, in 5-ITU-pretreated 32-day-old SE rats, a significantly higher AD threshold was demonstrated (*p* = 0.0057; F = 7.997) (Figure 2).

### 2.2. Duration of Hippocampal Afterdischarges

LiCl/pilocarpine SE resulted in a significantly shorter AD duration in 15-day-old rats after the first three stimulations (out of six) compared to age-matched controls (*p* = 0.0001–0.0381; F = 15.77–25.94). The AD duration at any stimulation intensity did not significantly differ between controls and LiCl/pilocarpine SE in 12, 18, 25, and 45-day-old rats. In contrast, a significant prolongation of AD duration was demonstrated in 32-day-old rats after SE compared with age-matched controls, which appeared after all stimulations (*p* = 0.0104–0.0350; F = 18.09–26.99) (Figure 2).

5-ITU led to significant changes in the AD duration in both the control and SE groups of animals. In control rats, a significant decrease in AD duration in 5-ITU-pretreated 12-day-old rats (*p* = 0.0014–0.0354; F = 11.31–18.96) and 15-day-old rats (*p* = 0.0008–0.0389; F = 8.284–16.98) was demonstrated. Similar significant changes in 12-day-old rats (*p* = 0.0019–0.0055; F = 9.273–12.88) and 15-day-old rats (*p* = 0.0008–0.0126; F = 7.056–11.99) were demonstrated between 5-ITU-pretreated control and SE rats. In the SE group, the decreased duration of ADs was shown in 5-ITU-pretreated 12-day-old rats (*p* = 0.001–0.0202; F = 7.213–10.24), 15-day-old rats (*p* = 0.00419; F = 18.93), and 32-day-old rats (*p* = 0.007–0.0014; F = 13.80–15.34) (Figure 2).

### 2.3. Analysis of Adenosine Kinase Isoforms

Naive animals tended to experience increased ADK-L isoform levels in the hippocampus compared to the youngest group (P7), but statistical significance was not reached (Figure 3).

The ADK-S isoform showed a stepwise increased expression during brain development. Statistically significant increases were demonstrated in 18-day-old rats (*p* = 0.0047; F = 19.02), 21-day-old rats (*p* = 0.0416; F = 14.92), 32-day-old rats (*p* = 0.0342; F = 18.03), and 45-day-old rats (*p* = 0.0280; F = 15.51) compared to the youngest group. The highest level of ADK-S isoform expression was demonstrated in 18-day-old rats and was significantly higher compared to 10-day-old rats (*p* = 0.0114; F = 17.32) and 12-day-old rats (*p* = 0.0199; F = 12.37) (Figure 3).

The ADK-L/ADK-S ratio revealed a statistically significant gradual decline in 12 day old rats (*p* = 0.0474; F = 1.251), 15 day old rats (*p* = 0.0124; F = 6.869), 18 day old rats (*p* = 0.0043; F = 638.7), 21 day old rats (*p* = 0.0043; F = 716.7), 25 day old rats (*p* = 0.0043; F = 686.3), 32 day old rats (*p* = 0.0041; F = 2257), and 45 day old rats (*p* = 0.0040; F = 2064) compared to the youngest (7 day old) group. In conclusion, the dominance of ADK-L isoform over ADK-S isoform expression demonstrated that the ADK-L/-S ratio declined stepwise during early brain development (Figure 3).

In SE rats, ADK-L isoform showed rapid downregulation 6 h after SE induction compared to appropriate age-matched controls (*p* = 0.021; F = 13.07). On the other hand, a significant upregulation of the ADK-S isoform in the same age group was revealed (*p* = 0.0425; F = 1.174). Both ADK-S and ADK-L isoforms exhibited a tendency toward increased expression during brain development after LiCl/pilocarpine SE, but the significant increase was found only in the ADK-L isoform in 15-day-old SE rats (*p* < 0.0001; F = 538.9) and 32-day-old SE rats (*p* = 0.0466; F = 174.72) compared to 12-day-old SE rats. The ADK-L/-S ratio showed a significant increase in 32-day-old SE rats compared to 15-day-old rats (*p* = 0.0103; F = 6.003) and 12-day-old rats SE rats (*p* = 0.0005; F = 112.4) and in 15-day-old SE rats compared to 12-day-old SE rats (*p* = 0.0005; F = 18.75) (Figure 4).

## 3. Discussion

The LiCl/pilocarpine model of status epilepticus (SE) is the most popular and widely used rodent model for the study of TLE [35]. The increased hippocampal excitability in TLE is expectable [43]. SE initiates a cascade of neuropathological, molecular, and genomic changes [44,45].

LiCl/pilocarpine SE induced in immature rats at the age of 12 days might lead to brain hyperexcitability [46] that may lead to spontaneous recurrent seizures, i.e., the development of epilepsy in adulthood [36,47]. These findings are supported by several studies in which exposure of the immature brain to clinically relevant epileptogenic conditions increased brain excitability [48,49]. LiCl/pilocarpine SE is principally responsible for hippocampal neuronal loss [50,51,52]. The apparent relationship between the severity of neuronal damage and the development of epilepsy has been demonstrated in adult rodents [53,54]. However, controversial findings on hippocampal alterations after LiCl/pilocarpine SE in immature rats have been published [55,56,57].

To understand the impact of the LiCl/pilocarpine SE of immature rats on hippocampal excitability, we used hippocampal Ads, a model of focal temporal seizures [29]. As expected, typical automatism wet dog shakes were observed, which can be taken as a sign of hippocampal involvement [58]. In immature rats, WDSs were not regularly present during the first two postnatal weeks but started to occur upon maturation [59]. The hippocampal AD characteristics such as duration, morphology, and thresholds have been investigated in epilepsy patients. The prolonged duration and decreased thresholds of Ads have been associated with increased risk of seizures in epilepsy patients [34]. Moreover, initial fast polyspikes (PSs) have been presented as the most common hippocampal AD morphology within the irritative zone [34,60], and AD thresholds for hippocampal stimulation were significantly lower in epilepsy patients [60]. The duration and threshold of ADs were studied in our study.

Our experiments found age-related changes in the duration of hippocampal ADs. LiCl/pilocarpine SE led to a marked decrease in hippocampal excitability within 3 days of SE. In contrast, a significant increase in hippocampal excitability was demonstrated 20 days after SE. These observations are similar to those seen in previous studies using the pilocarpine SE model in immature rats [47,61].

It was suggested that epilepsy development in the immature brain may not require neuronal death [62], and the inhibition of neuronal loss in the hippocampus was not sufficient to prevent the development of epilepsy [63]. The mechanisms through which LiCl/pilocarpine SE increases hippocampal excitability in immature rats are not clear, and the risk of seizures later in life is not fully understood. Roles for excitatory receptors and transporters [64], synaptic reorganization [65], oxidative stress, and mitochondrial dysfunction [66], in addition to the contribution of astrocytes, have been suggested [67] after pilocarpine-induced SE in immature rats.

Astrogliosis with brain hyperexcitability has been associated with the development of epilepsy in adult and immature rats [67,68,69,70,71]. Astrocytes are tightly involved in the metabolism of the inhibitory neuromodulator and endogenous anticonvulsant adenosine [72]. The anticonvulsant effect of adenosine has been reported in models of status epilepticus, including the LiCl/pilocarpine model [73,74]. Adenosine kinase (ADK) has been shown to critically regulate extracellular adenosine levels in the brain [40,75]. The upregulation of ADK corresponds with neuronal hyperexcitability by decreasing the levels of endogenous adenosine [76,77], as well as contributes to increased DNA methylation.

Moreover, two isoforms of ADK with different subcellular localizations have been identified [18]. Upregulation of both ADK isoforms leads to increased DNA methylation, but the ADK-L isoform appears to be more effective in the regulation of DNA methylation [19]. During early postnatal brain development, a dramatic switch has been described from neuronal to glial localization and from ADK-L to ADK-S isoform expression [18,23]. In rats, these changes occur during the first 3 weeks of life [23].

In this study, we demonstrated changes in ADK isoform (ADK-L and ADK-S) expression during normal development of the hippocampus and, for the first time, changes after LiCl/pilocarpine SE. We demonstrated the stepwise increase in ADK-S isoform expression during hippocampal development. Levels in 18 day old and older rats (with the exception of 25-day-old rats) were significantly higher compared to the levels in the youngest animals. In contrast, changes in the ADK-L isoform during brain development did not reach the level of significance, but a decreasing ADK-L/-S ratio showed the increasing dominance of ADK-S isoform during brain development in this study. A similar shift in the expression ratio to ADK-S dominance over ADK-L during early brain development in mice was recently published [25]. A marked decrease in the ADK-L/-S ratio of the cerebrum in 7- compared to 15- and 21-day-old mice was demonstrated. We demonstrated similar results in the same age groups of rats, as well as similar changes in the ADK-L/-S ratio of the hippocampus in 7-day-old rats compared to other representatives of the perinatal (10- and 12-day-old rats), infantile (18-day-old rats), juvenile (25- and 32-day-old rats), and adolescent (45-day-old rats) groups. These data correspond with previously published studies that described opposite shifts in isoform expression in different cell types, with downregulation of the ADK-L isoform in neurons and upregulation of the short isoform in astrocytes during brain development [24]. The major shifts in hippocampal ADK expression occur toward the end of the first postnatal week [24].

Additionally, we presented the changes in ADK isoform expression after LiCl/pilocarpine SE in the early stages of hippocampal development. The overexpression of ADK was demonstrated under different conditions of acute brain injury after the latent phase preceding the development of epilepsy [78,79]. ADK is known to be rapidly downregulated after SE induction [42,80]. ADK expression throughout the hippocampal formation shows acute reduction followed by increases in ADK expression in the systemic kainic acid-induced SE in mice [19,42]. Previously described ADK downregulation likely represents an attempt to increase levels of the anticonvulsant adenosine after an acute insult [81], but the changes in ADK isoform expression in rats with SE during early brain development have not yet been elucidated. Diverse insults, such as SE, trigger astroglial activation [68]. Astrogliosis may, thus, suggest the dysregulation of ADK expression and might contribute to recurrent seizure activity.

In our study, a rapid decrease in the ADK-L isoform and an increase in ADK-S isoform 2 h after LiCl/pilocarpine SE induction (12-day-old rats) compared to controls were shown. With the latency after SE, only the overexpression of ADK-L isoform in 15- and 32-day-old rats compared to 12-day-old SE rats was revealed. ADK-S isoform in 15- and 32-day-old rats did not reach a level of significance compared to controls or SE rats. In the SE group, the representatives of 15- and 32-day-old rats were chosen due to the opposite effect of SE on the hippocampal ADs.

The ADK-L/-S ratio corresponded with the increasing dominance of the ADK-L isoform in 32-day-old SE rats compared to younger SE groups. The ADK-S isoform has been considered to be involved in the regulation of brain excitability by the modulation of adenosine extracellular tone [8]. However, the ADK-L isoform has been more pronounced as an epigenetic regulator. Increased levels of ADK-L isoform drive increased DNA methylation [19]. These results may support the hypothesis that the ADK-L isoform is considered to be a major contributor to the development of epilepsy and progression [20,82]. The changes in hippocampal excitability after LiCl/pilocarpine SE may, thus, be partly explained by dysregulation of each ADK isoform expression.

Inhibition of ADK leads to rapid increases in adenosine levels [17,83]. In various models, pharmacological inhibition of ADK suppresses seizures [15,42,83]. The most potent selective inhibitor of ADK, 5-iodotubercidin [15], reduced synaptic transmission in the hippocampus by increasing adenosine levels in the brain [13,41] and may foster seizure suppression following bicuculline-induced seizures [84] or kainic acid SE in adult rats [17]. Nevertheless, the effect of selective ADK inhibitors on immature rats has not been reported.

5-ITU was studied in developmental stages corresponding to the human perinatal, preschool (infantile), and school-age (juvenile) periods. Hippocampal ADs were suppressed by the adenosine kinase inhibitor 5-ITU only in 32-day-old rats, 20 days after LiCl/pilocarpine SE. The prolongation of hippocampal ADs and decreased threshold of the stimulation current intensity suggest that the ADK-L isoform plays an important role in hippocampal excitability after LiCl/pilocarpine SE. This possibility should be studied in the future.

Despite intense research into the development of antiepileptic drugs, no clinically viable pharmacological therapy exists to prevent the development of epilepsy after SE during early brain development. Inhibitors of ADK isoforms might be a promising approach. Our results support the important role of ADK isoforms modifications after LiCl/pilocarpine SE on the excitability of the developing hippocampus, but the exact mechanisms to verify the hypothesis remain to be experimentally scrutinized. Direct detection of adenosine concentration in the hippocampus during the hippocampal ADs may represent the possibility to confirm the hypothesis.

Measuring direct changes of adenosine is particularly challenging. Adenosine’s half-life of the order of seconds [85] and the constant cellular release and enzyme degradation [86] represent a high risk of measurement error for methods such as microdialysis. The conventional detection methods for adenosine, such as capillary electrophoresis and high-performance liquid chromatography (HPLC), are not suitable for in vivo real-time on-site detection [87]. Another limitation of the study is the technical impossibility of long-term monitoring of spontaneous seizures and hippocampal ADs in the same cohort of immature rats due to some developmental aspects. An incomplete skull ossification and premature synostosis with duracrol fixation leading to brain damage [88], a high risk of electrode removal by the mother before weaning or restriction of rats’ normal behavior, and ictal semiology during seizures due to heavy monitoring equipment represent the major issues that have to be considered.

## 4. Materials and Methods

### 4.1. Animals

The procedures involving animals were conducted according to the ARRIVE guidelines (https://www.nc3rs.org.uk/arrive-guidelines, accessed on 29 January 2022) in compliance with national (Act No. 246/1992 Coll.) and international laws and policies (EU Directive 2010/63/EU for animal experiments) and the National Institutes of Health Guide for the Care and Use of Laboratory Animals (NIH Publications No. 8023, revised 1978). The experimental protocol was approved by the Ethical Committee of the Czech Academy of Sciences (Approval No. 15/2018).

Experiments were performed using 266 male albino Wistar rats (bred at the Institute of Physiology, Czech Academy of Sciences, Prague, Czech Republic) at postnatal (P) days P7, P10, P12, P15, P18, P21, P25, P32, and P45. The day of birth was counted as P0, and weaning took place at P21. Animals were housed in a controlled environment (12:12 h light/dark cycle, temperature 22 ± 1 °C, humidity 50–60%) with ad libitum access to food and water.

### 4.2. Status Epilepticus

At the age of 11 days (P11), all animals were injected with lithium chloride (127 mg/kg i.p.). At P12 (i.e., 24 h later), pilocarpine at a dose of 40 mg/kg was administered i.p. to four or six rats in each nest (total of 134 animals). The control rats (lithium paraldehyde (LiPARA) group, four to six out of 10 in the nest, a total of 132 animals) received saline in the same volume (4 mL/kg) instead of pilocarpine. After pilocarpine administration, the behavior of the rats was closely monitored for approximately 2 h to evaluate the onset time of stage 4 seizure, SE, severity, and mortality. SE was defined as a continuous motor seizure at stage 4 (bilateral forelimb clonus with rearing), stage 5 (loss of balance, rearing, and falling), and stage 6 (severe tonic–clonic seizures, with a loss of righting reflexes) [89]. In this study, 112 rats that showed severe tonic–clonic seizures (stage 6) were included: 10 rats that did not develop SE, four rats at stage 4, and six rats at stage 5 were excluded. A total of two rats died during SE. After 90 min of sustained convulsive activity (or at corresponding times in controls), paraldehyde (0.07 mL/kg i.p.) was injected to interrupt seizures and decrease mortality. Control rat pups received paraldehyde (0.07 mL/kg i.p.) 90 min after the saline administration. The animals spent approximately 2 h in isolation and were then returned to their mothers. The body temperature of P12 (as well as P15) rat pups was maintained during the whole experiment by means of an electric plate heated to 34 °C, i.e., the temperature in the nest.

### 4.3. Surgery

Surgery was performed under isoflurane 0.5–2% anesthesia. A deep hippocampal stimulation electrode (Plastics One, Roanoke, VA, USA) was implanted stereotaxically into the right dorsal hippocampus, and a recording electrode was implanted into the left dorsal hippocampus at coordinates AP −3.0 mm, L +2.8 mm, D +3.0 mm for young adult rats; the coordinates were recalculated for immature animals on the basis of the bregma–lambda distance. After the stimulation procedures, animals were sacrificed, and the location of the electrodes was histologically verified in Nissl-stained sections of the hippocampus (Figure 5). Two flat silver recording electrodes were placed epidurally over the sensorimotor cortex of the left and right hemispheres at coordinates AP +1 mm, L +2 mm. Reference and ground electrodes were placed over the cerebellum. Electrodes were connected to a six-pin connector and fixed to the skull with fast-curing dental acrylic (Duracrol©, Dental, Prague, Czech Republic). Implantation of electrodes took less than 30 min, at which time the isoflurane anesthesia was discontinued. The animals were allowed to recover for at least 1 h before the experimental procedure started. The 12-day-old SE rats were allowed to recover for at least 6 h after the termination of convulsive SE. Rat pups with immature thermoregulation (up to the end of the third postnatal week) were placed on an electric pad heated to 34 °C during the recovery period, as well as during the experiment. Animals with stereotaxically implanted electrodes were not included in the Western blot analysis.

### 4.4. Stimulation Procedure, Afterdischarges and Threshold Intensity

Animals were placed individually into plastic boxes and connected to the amplifier and stimulator. An isolated pulse stimulator (Model 2100, A-M Systems, Sequim, WA, USA) with constant current output was used. Hippocampal ADs were elicited by a series of biphasic 1 ms pulses applied for 2 s at a 60 Hz frequency. With the threshold intensities of stimulation current, the total duration of electrographic ADs was measured. The electrographic onset of ADs was characterized by initial fast polyspike (PS) and large delta and/or sharp theta waves [32], usually followed by a pattern of low-amplitude fast oscillation [90]. The end of fast oscillation was selected as the termination of the ADs (Figure 6).

Wet dog shakes (WDSs) were observed at the end of an electroclinical seizure or the end of ADs. WDSs are associated with stereotyped behavior (automatisms) characteristic of seizures originating in limbic structures [58]. WDSs were not regularly present during the first two postnatal weeks but started to occur upon maturation [59].

The suprathreshold intensity was estimated using current intensities from 0.2 mA to 2.0 mA and was used for all sessions. Six series of 2 s stimulations were applied at 20 min intervals. The stimulation procedure was performed at least 2 h after the electrode implantation. Each rat was connected to the amplifier and monitored for at least 20 min before the stimulation procedure started to avoid the possible interference of spontaneous seizures with ADs.

### 4.5. Recording

The TDT Open Project Program (Tucker-Davis Technologies, Alachua, FL, USA) was used to record all electrophysiologic signals. All obtained signals were amplified (Pentusa Base Station, Tucker-Davis Technologies, Alachua, FL, USA) and digitized at 1 kHz.

### 4.6. Drugs

The ADK inhibitor 5-iodotubercidin (5-ITU) purchased from Tocris Biosciences (#1745) (Bristol, UK), was administered intraperitoneally at a dose of 3.1 mg/kg. The most potent ADK inhibitors compound showed potencies within the range of 0.3–7.1 mg/kg [83], but the details of 5-ITU pharmacokinetics are not fully clarified.

The dose was selected according to Gorter et al. [80]. According to data obtained from the stimulation series, three age groups, representing the perinatal, infantile, and juvenile developmental periods, were used for 5-ITU administration: P12, P15, and P32. The experimental (LiCl/pilocarpine SE) group comprised 7–8 animals. 5-ITU was put in suspension with a drop of Tween-80 and diluted with saline immediately before the experiment to obtain a final concentration of 3.1 mg/mL. The control (lithium paraldehyde (LiPAR)) group was used for comparison. Animals were injected with 5-ITU or saline 10 min before the start of the stimulation procedure.

### 4.7. Western Blot Analysis

Western blot analysis was performed to detect changes in ADK isoforms (ADK-L and ADK-S) in nine groups (7, 10, 12, 15, 18, 21, 25, 32, and 45-day-old rats) of naïve rats (without LiCl administration) and in three age groups in rats after SE (12, 15, and 32 day old rats). Hippocampal tissue from 48 intact rats (four animals/group in naive; four animals/group in SE) was collected. Animals with stereotaxically implanted electrodes were not included. The tissue was frozen and stored at −80 °C before analysis (Figure 7). All mixed samples were prepared from six animals using a glass homogenizer with a power-driven Teflon pestle (Helidolph–RZR2021) with 10 mM PBS (pH 7.4) at a 1:4 ratio and protease inhibitor cocktail (# P8340, Sigma-Aldrich, St. Louis, MO, USA). The homogenates were centrifuged (#120951, Sigma-Aldrich 2–16 PK, Sigma-Aldrich, St. Louis, MO, USA) at 1000× *g* for 10 min at 4 °C, and the supernatant was collected. A small volume of the hippocampal lysate was used for quantification of the protein concentration by Lowry’s method [91] with Peterson’s modification [92]. Before electrophoresis, the samples were mixed in a 1:2 ratio with Laemmli loading buffer (#161-0737, Bio-Rad, Hercules, CA, USA) and heated for 20 min at 70 °C. Stain-free gradient gels (#567-8104, Bio-Rad, Hercules, CA, USA) were used for protein separation and protein labeling by the binding of trihalo compound to tryptophan residues. After electrophoresis (300 V, 270 mA, 23 min), all gels were activated and visualized by UV light for 5 min by a ChemiDoc™ Touch Imaging System (Bio-Rad, Hercules, CA, USA). The samples were subsequently transferred to nitrocellulose membranes (#170-4271, Bio-Rad, Hercules, CA, USA) using a Trans-blot Turbo apparatus (Bio-Rad, Hercules, CA, USA) (Figure 2). The quality of transfer and volume of protein on the membrane was determined by a ChemiDoc™ Touch Imaging System (Bio-Rad). Membranes were blocked in 5% nonfat milk in Tris-buffered saline (TBS) for 1 h at room temperature and were then incubated overnight at 6 °C with primary antibodies against ADK (1:3000; PA5-27399, Thermo Fisher Scientific, Waltham, MA, USA). The following day, membranes were washed for 3× 10 min in TBS, incubated for 1 h at room temperature with secondary antibody (1:30,000; #211-032-171, Jackson ImmunoResearch Laboratories, West Grove, PA, USA), and then washed again in TBS, as described above. The chemiluminescent substrate (Supersignal WestFemto, #34096, Thermo Fisher Scientific, Waltham, MA, USA) was used for visualization of the protein with the ChemiDoc™. Stain-free images of total protein were used to normalize the target protein as a loading control. Western blotting normalization with stain-free technology is comparable to other total protein staining methods (PonceauS, Coomassie Blue, etc.) [93].

The analysis of membrane images and the optimal exposure time were performed using the ImageLab 6.1, (Bio-Rad, Hercules, CA, USA). The optimal exposure time was used to visualize very faint bands and eliminate the overexposed prominent bands. The obtained value of the chemiluminescent signal of the target antibody (ADK antibody) was normalized to the total protein detected in each line, i.e., multiplied by a normalization factor inversely proportional to the protein content. For each line, the normalization factor was calculated individually using the first line as a standard with a normalization factor of 1.0. The specific normalization factor was equal to the ratio of the total signal intensity obtained in the first-line stain-free image to the total signal intensity in the target line. The background signal was subtracted from the total signal intensity in each line. This calculation was performed semi-automatically by the software with the need to accurately identify the areas (lines and bands) being compared.

### 4.8. Statistics

Statistical evaluation of the electrophysiologic data was performed with Prism (version 8.0; GraphPad, La Jolla, CA, USA). This program started with a test of the distribution of the data and recommended a parametric or nonparametric test on the basis of the results.

The duration of ADs and threshold intensities in individual ages in control vs. SE groups were evaluated with Sidak’s test and two-way ANOVA RM (or mixed model) corrected with multiple comparisons. Hippocampal ADs and threshold intensities in particular ages in control or SE groups were compared using the *t*-test. A *p*-value <0.05 was considered statistically significant.

Statistical analysis and graphs of normalized intensity units in Western blot analysis were performed again with Prism (version 8.0; GraphPad, La Jolla, CA, USA). Differences between the nine age groups were evaluated with one-way ANOVA, and the *t*-test was subsequently used to determine the statistical significance (*p* < 0.05 or *p* < 0.001). Data are presented as the optical density values ± SEM.

The ratio of ADK-L to ADK-S isoform expression was performed by quantitative analysis of the Western blot bands using the ImageJ software (version 1.52; ImageJ, WI, USA) and expressed as the ratio of optical densities of ADK-L/-S bands.

## 5. Conclusions

The development of ADK isoforms in the immature brain and changes in the model of LiCl/pilocarpine SE were demonstrated. We detected changes in ADK isoform expression in the model of LiCl/pilocarpine SE compared to intact brain development. The changes are age-dependent and may correlate with increased hippocampal excitability after LiCl/pilocarpine SE. The increasing expression of the ADK-L isoform after SE may contribute to the markedly increased excitability, as demonstrated by the thresholds and total duration of AD in our experiments. 5-ITU led to a marked decrease in hippocampal excitability. Thus, inhibitors of ADK might be useful as antiepileptic drugs in neurology.

## Figures and Tables

**Figure 1 ijms-23-02510-f001:**
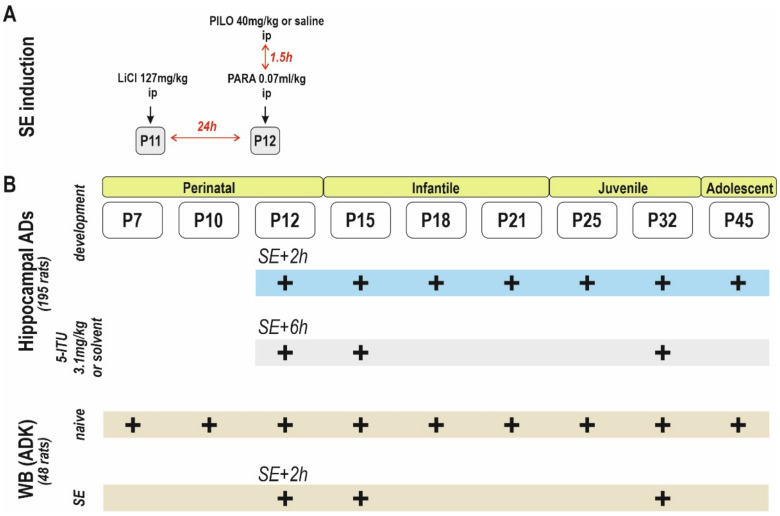
Schematic figure illustrating the experimental design of the study. (**A**) LiCl/pilocarpine induction. (**B**) From top to bottom: hippocampal ADs elicited during brain development and Western blot (WB) analysis. For details, see inset.

**Figure 2 ijms-23-02510-f002:**
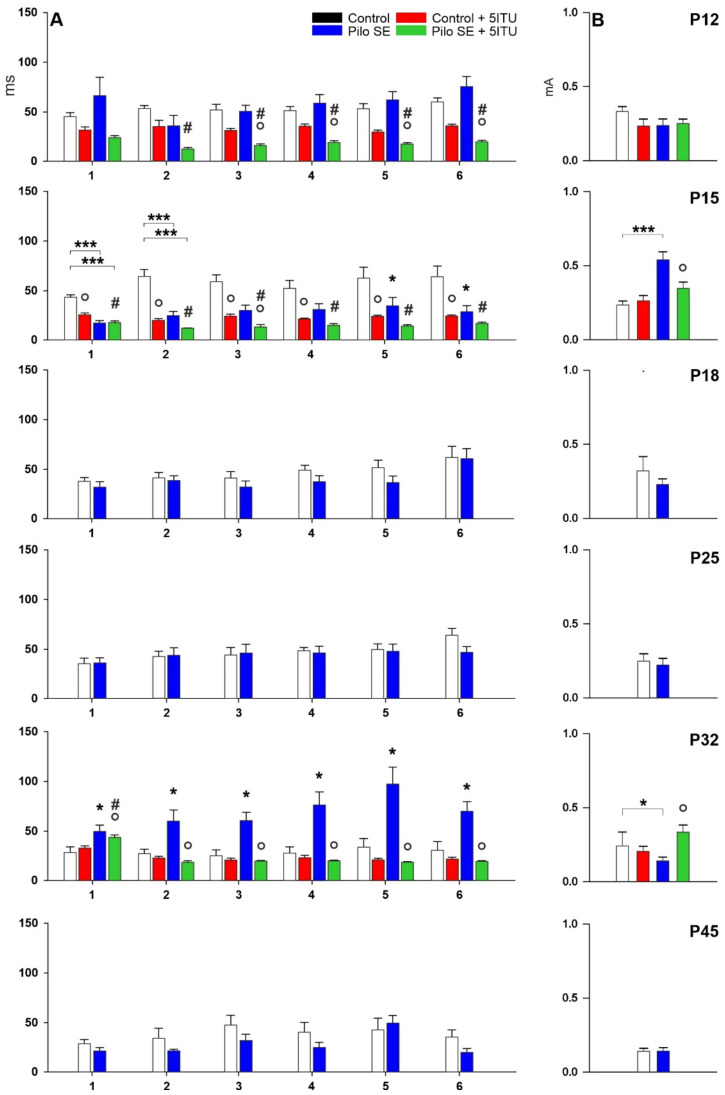
Effects of LiCl/pilocarpine SE, the effect of adenosine kinase inhibitor 5-ITU on the total duration of hippocampal ADs (mean + SEM), and the threshold intensity for elicitation of hippocampal afterdischarge (mean + SEM) in seven age groups. From left to right: (**A**) duration of hippocampal ADs, (**B**) threshold intensity. Black color—control rats, red color—control animals given 3.1 mg/kg of 5-ITU, blue color—LiCl/pilocarpine SE rats, green color—LiCl/pilocarpine SE rats given 3.1 mg/kg of 5-ITU (see inset). From top to bottom: 12-, 15-, 18-, 25-, 32-, and 45-day-old rats. (**A**) Abscissae: number of stimulations; ordinates: the total duration of ADs in seconds. One or three asterisks denote significant differences from corresponding control ADs for * *p* < 0.05 or *** *p* < 0.001, respectively. Circles denote significant differences from corresponding ADs in 5-ITU-pretreated rats compared to age-matched controls or SE rats for * *p* < 0.05. Hashtags denote significant changes from between 5-ITU-pretreated SE rats compared to age-matched controls for * *p* < 0.05. (**B**) Ordinates: intensity of stimulation in mA. One or three asterisks denote a significant difference from corresponding control ADs for * *p* < 0.05 or *** *p* < 0.001, respectively. Circles denote significant differences from corresponding thresholds in 5-ITU-pretreated rats compared to age-matched SE rats for * *p* < 0.05.

**Figure 3 ijms-23-02510-f003:**
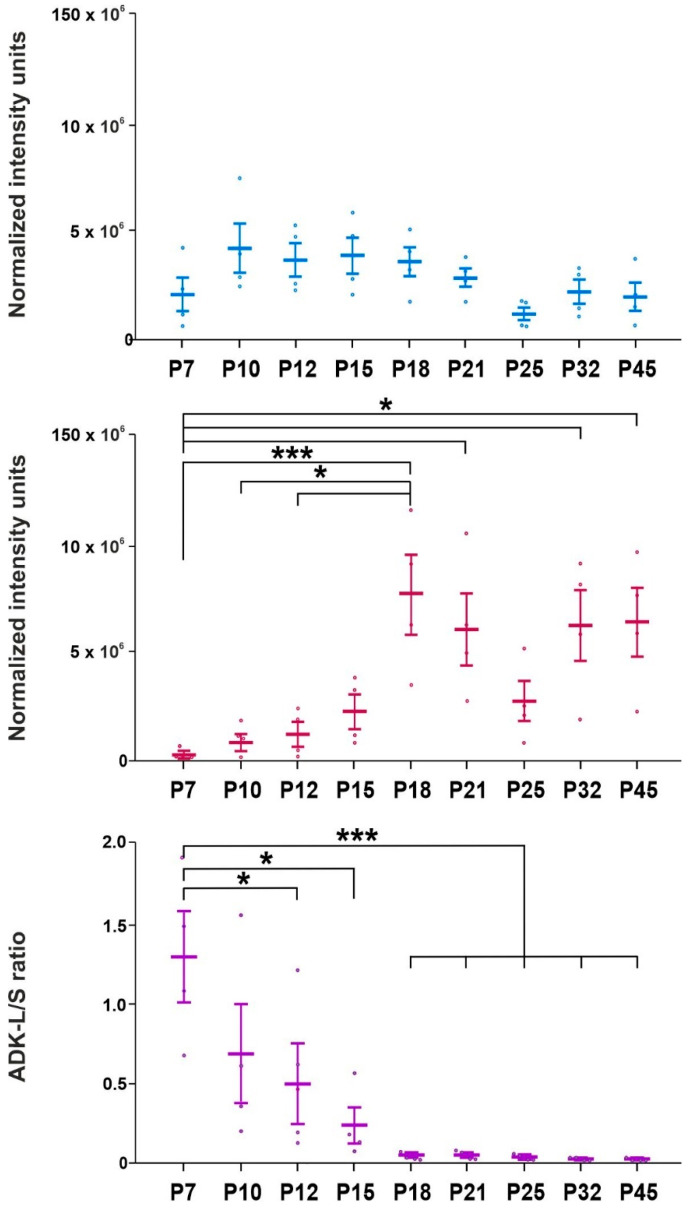
Analysis of ADK isoforms in controls, ADK-L and ADK-S (mean gray value + SEM), in nine age groups. From top to bottom: ADK-L in controls, ADK-S in controls, ADK-L/-S ratio in controls. Abscissae from left to right: 7-, 10-, 12-, 15-, 18-, 21-, 25-, 32-, and 45-day-old rats; ordinates: normalized intensity units or ratio (see inset). One or three asterisks denote a significant difference for * *p* < 0.01 or *** *p* < 0.001, respectively (details, see inset).

**Figure 4 ijms-23-02510-f004:**
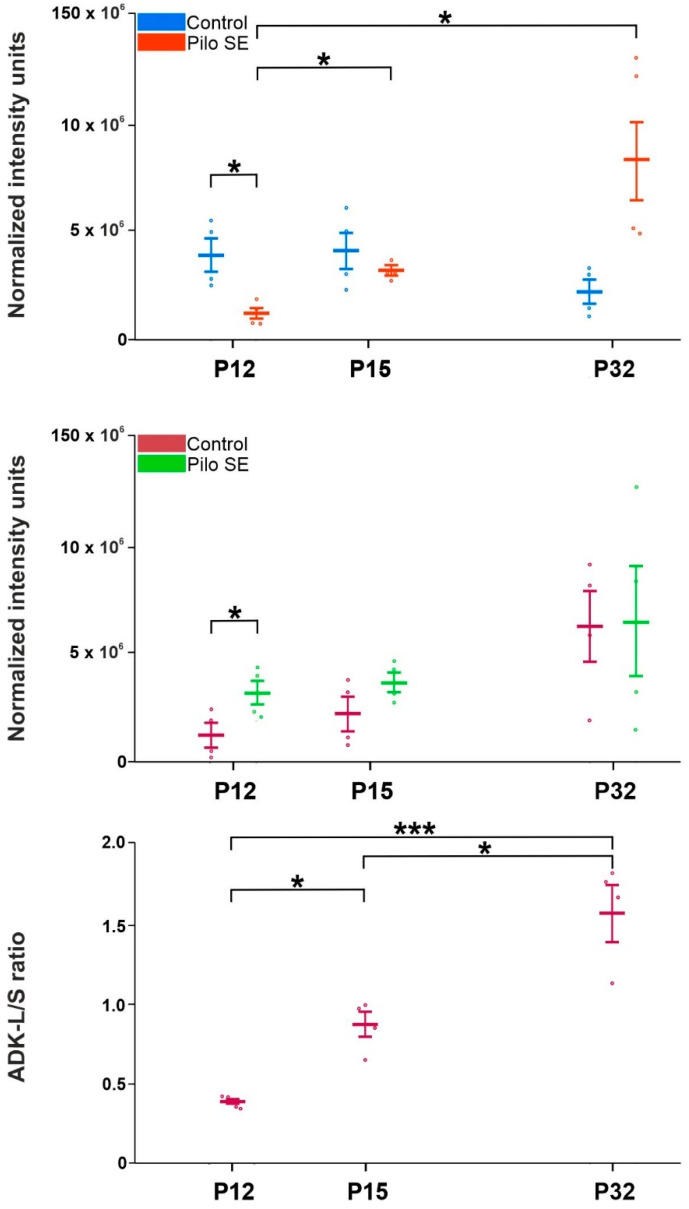
Analysis of ADK isoforms in control and LiCl/pilocarpine SE rats, ADK-L and ADK-S (mean gray value + SEM), in three age groups. From top to bottom: ADK-L in LiCl/pilocarpine SE, ADK-S in LiCl/pilocarpine SE, ADK-L/-S ratio in LiCl/pilocarpine SE. Abscissae from left to right: 12-, 15-, and 32-day-old rats in controls and LiCl/pilocarpine SE (see inset); ordinates: normalized intensity units or ratio (see inset). One or three asterisks denote a significant difference for * *p* < 0.01 or *** *p* < 0.001, respectively (details, see inset).

**Figure 5 ijms-23-02510-f005:**
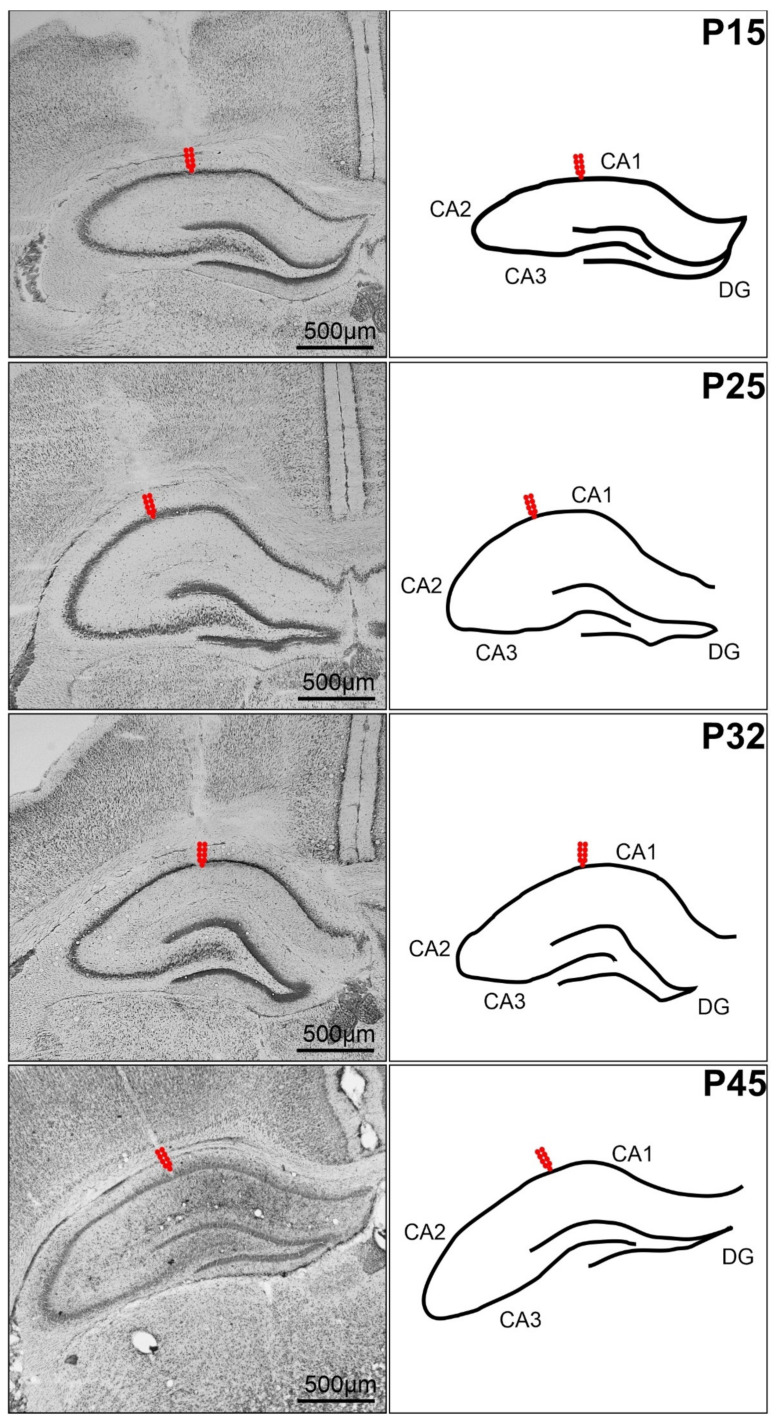
Coronal sections of rat hippocampus after electrical stimulation of the CA1 regions. From left to right: in Nissl-stained sections and in schematic imagery. From top to bottom: 15, 25, 32, and 45-day-old rats. Arrows show the location of the registration electrodes. For scalebar, see inset.

**Figure 6 ijms-23-02510-f006:**
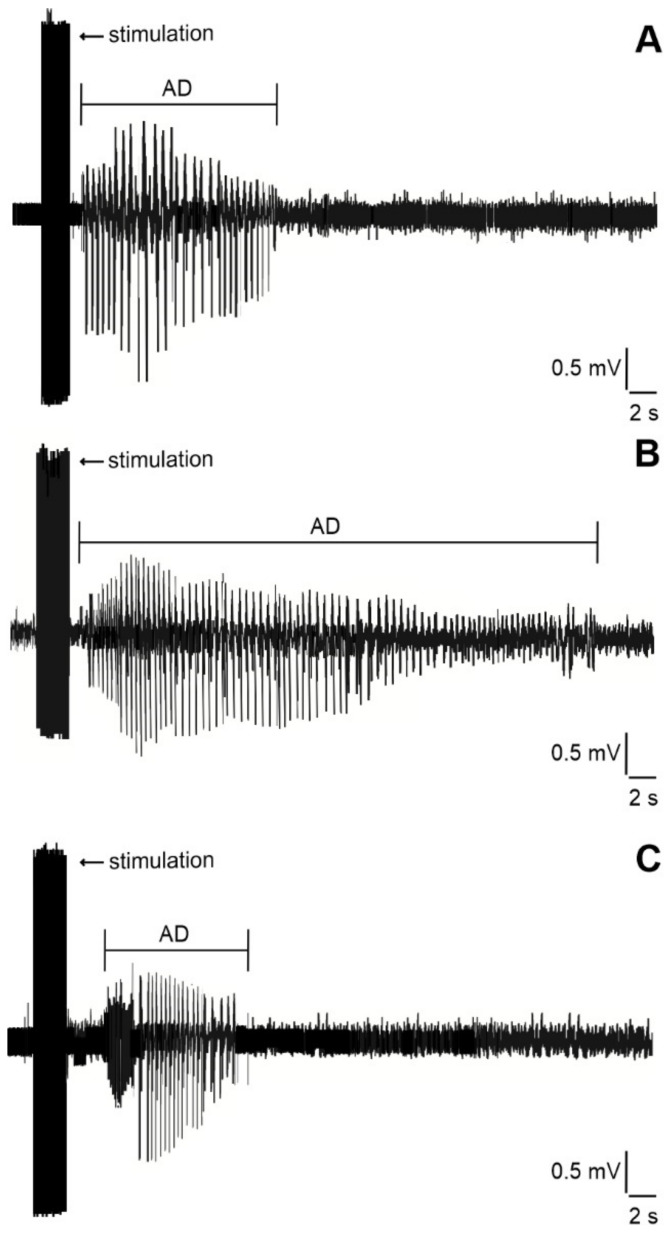
Hippocampal ADs elicited by electrical stimulation in 32-day-old rats accompanied by epileptic automatisms (wet dog shakes). From top to bottom: (**A**) in control animal, (**B**) in LiCl/pilocarpine SE rat, (**C**) in LiCl/pilocarpine SE rats after the administration of 5-ITU at a dose of 3.1 mg/kg (see inset).

**Figure 7 ijms-23-02510-f007:**
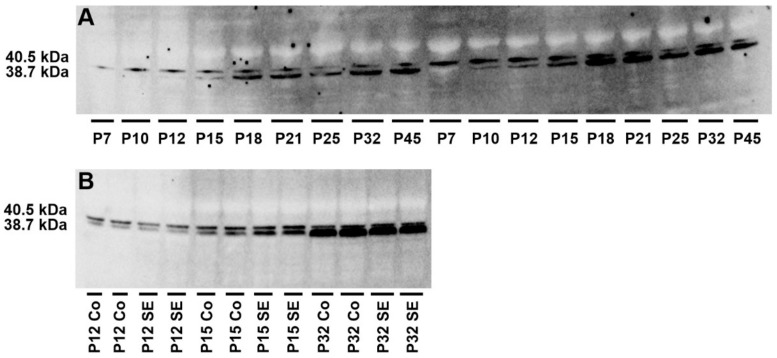
Western blotting of hippocampal homogenates showing the density of ADK isoforms–ADK-S (38.7 kDa) and ADK-L (40.5 kDa) during brain development in control rats (**A**) and corresponding age-matched LiCl/pilocarpine SE rats (**B**). (**A**) Abscissae from left to right: 7, 10, 12, 15, 18, 21, 25, 32, and 45-day-old rats in identical replicates. (**B**) Abscissae from left to right: 12-, 15-, and 32-day-old rats in identical replicates.

## Data Availability

All data generated or analysed during this study are included in this published article.

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
