# Peer review of "Adenosine Kinase Isoforms in the Developing Rat Hippocampus after LiCl/Pilocarpine Status Epilepticus"

_ijms, 2022, doi:10.3390/ijms23052510_

Round 1
Reviewer 1 Report
Fabera Petr et al. investigated changes in the expression of two adenosine kinase (ADK) isoforms in the developing rat hippocampus. The authors also showed that early SE in the lithium-pilocarpine model resulted in decreased expression of ADK-L isoform in 12-day-old rats. In contrast, ADK-S isoform expression increased at the same age compared to naive rats. The authors suggest that these changes may affect the excitability of the dorsal hippocampus after SE. After SE, excitability in the hippocampus decreases (3 days after SE) and then increases (20 days after SE). These results are new and may be of interest to researchers studying epileptogenesis.
Comments
- The causal relationship between excitability changes and ADK/isoform ADK expression is not explicitly shown. Therefore, the Title of the article may be somewhat misleading. It is likely that the Running title more accurately reflects the article's content. The limitations of the method used to detect the relationship between excitability and ADK expression should be discussed.
- Direct measurement of the amount of adenosine in the brains of control, post-SE rats and when exposed to 5-iodotubercidin, as well as correlation with the duration/threshold of ADs, would be more convincing evidence.
- Fig. 4. Is there a need to show data for each of the six series? If so, please explain to the reader the intent. For example, what changes in the course of the series should the reader see? Otherwise, it is better to show the average data for the six series because a large number of small points makes it difficult to distinguish the main differences.
- Figs. 5 and 6. Normalized intensity units. Please change the scale to decrease the number of digits (×106).
- Fig. 7: Some bands look overloaded (P32 SE), which may lead to measurement errors (out of the linear range). Have you used any criteria to avoid this kind of error?
Author Response
Reviewer 1:
1/ The causal relationship between excitability changes and ADK/isoform ADK expression is not explicitly shown. Therefore, the Title of the article may be somewhat misleading. It is likely that the Running title more accurately reflects the article's content. The limitations of the method used to detect the relationship between excitability and ADK expression should be discussed.
Direct measurement of the amount of adenosine in the brains of control, post-SE rats, and when exposed to 5-iodotubercidin, as well as correlation with the duration/threshold of ADs, would be more convincing evidence.
We agree that the title of the manuscript is misleading. Thus, we have decided to change it. The new title is “Adenosine kinase isoforms in the developing rat hippocampus after LiCl-pilocarpine status epilepticus.“ (Page 1, lines 1-2)
Despite the results supporting the role of ADK isoforms expression after LiCl-pilocarpine SE on the excitability of the developing hippocampus, the exact mechanisms of ADK isoforms remain to be experimentally scrutinized. Direct detection of adenosine concentration in the hippocampus during and after the hippocampal ADs may represent the possibility to confirm the hypothesis. But measuring immediate changes of adenosine is particularly challenging. Adenosine half-life of the order of seconds (Jarvis et al., 2019) and constant cellular release and enzyme degradation (Lietsche et al., 2016) represent for methods such as microdialysis a high risk of measurement error. Other conventional detection methods for adenosine, such as capillary electrophoresis and high-performance liquid chromatography (HPLC), are not suitable for real-time on-site detection (Gaudin et al., 2015). (Page 16, lines 1-11)
Additionally, experimental artifacts due to implantation of electrodes and probes caused by damaging cell membranes on insertion, therefore lead to extremely poor signal noise ratios. Experiments with ADK isoform transgenic/knockout mice and selective ADK isoform inhibitors would be another promising approach.
2/ Fig. 4. Is there a need to show data for each of the six series? If so, please explain to the reader the intent. For example, what changes in the course of the series should the reader see? Otherwise, it is better to show the average data for the six series because a large number of minor points makes it difficult to distinguish the main differences.
We have presented data of the complete six stimulation series, but we have changed the graph presentation to emphasize the main differences (see Fig 4). The AD characteristics such as duration and threshold after stimulation have been investigated in epilepsy patients, the prolonged duration and decreased thresholds are associated with an increased risk of seizures in epilepsy patients (Gollwitzer et al., 2018). (Page 13, lines 7)
Moreover, the duration and threshold after each stimulation demonstrate the effect of 5-ITU from the administration, effective concentration (ED50) to the elimination of the drug. The details of 5-ITU pharmacokinetics are not fully clarified. (Page 8, lines 14-15)
3/ Figs. 5 and 6. Normalized intensity units. Please change the scale to decrease the number of digits (×106).
The scales have been changed based on the recommendation (see Fig. 5 and 6.)
4/ Fig. 7: Some bands look overloaded (P32 SE), which may lead to measurement errors (out of the linear range). Have you used any criteria to avoid this kind of error?
The optimal exposure time was automatically performed by the ImageLab program (BioRad; https://www.manualsdir.com/manuals/600179/bio-rad-gel-doc-ez-system-criterion-stain-free-tris-hcl-gels.html?page=50). The longest possible exposure time was used to visualize very faint bands (i.e., P7-P12). On the other hand, the longer exposure time may cause overexposure of more prominent bands that were not qualified. The ImageLab program marked these overexposed images in red. (Page 9, lines 27-29)
We can provide examples of band detection in P12 Co, P15 SE, and P32 SE.
P12 Co - see attached file
P15 SE - see attached file
P32 SE - see attached file
References:
Jarvis, M.F. Therapeutic potential of adenosine kinase inhibition-Revisited. Pharmacol Res Perspect. (2019) 22, 7(4):e00506. DOI: 10.1002/prp2.506
Lietsche, J.; Imran, I.; Klein, J. Extracellular levels of ATP and acetylcholine during lithium-pilocarpine induced status epilepticus in rats. Neurosci Lett. (2016) 12, 611:69-73. DOI: 10.1016/j.neulet.2015.11.028
Gollwitzer, S.; Hopfengartner, R.; Rossler, K.; Muller, T.; Olmes, D.G.; Lang, J.; Kohn, J.; Onugoren, M.D.; Heyne, J.; Schwab, S.; Hamer, H.M. Afterdischarges elicited by cortical electric stimulation in humans: When do they occur and what do they mean? Epilepsy Behav (2018) 87, 173–179. DOI: 10.1016/j.yebeh.2018.09.007
Gaudin, A.; Lepetre-Mouelhi, S.; Mougin, J.; Parrod, M.; Pieters, G.; Garcia-Argote, S.; Loreau, O.; Goncalves, J.; Chacun, H.; Courbebaisse, Y.; Clayette, P.; Desmaele, D.; Rousseau, B.; Andrieux, K.; Couvreur, P. Pharmacokinetics, biodistribution and metabolism of squalenoyl adenosine nanoparticles in mice using dual radio-labeling and radio-HPLC analysis. J Control Release. (2015) 28, 212: 50-8. DOI: 10.1016/j.jconrel.2015.06.016

Reviewer 2 Report
The authors have written an interesting paper about age-related differences in hippocampal excitability after SE, which might correspond to the de-velopment of ADK isoform levels in the hippocampus.
However, I have one minor complaint:
Abbreviation shouldn't be written in the title, i.e., "SE". Furthermore, abbreviation “SE” must be introduced in the abstract first, as well at the beginning of the introduction part.
Author Response
Reviewer 2:
1/ Abbreviation shouldn't be written in the title, i.e., "SE". Furthermore, abbreviation “SE” must be introduced in the abstract first, as well at the beginning of the introduction part.
Based on the comments of Reviewer 1 we have decided to change the title of the manuscript. The new title “Adenosine kinase isoforms in the developing rat hippocampus after LiCl-pilocarpine status epilepticus“ has been introduced. Abbreviations in the title as well as at the beginning of the abstract and introduction have been introduced (Page 1, lines 1-2; Page 3, line 2; Page 5, line 9).